# Active label cleaning for improved dataset quality under resource constraints

Mélanie Bernhardt[1,3], Daniel C. Castro[1,3], Ryutaro Tanno[1], Anton Schwaighofer[1], Kerem C. Tezcan[1],
Miguel Monteiro [1], Shruthi Bannur[1], Matthew P. Lungren[2], Aditya Nori[1], Ben Glocker [1],
Javier Alvarez-Valle [1] & Ozan Oktay [1✉]

Imperfections in data annotation, known as label noise, are detrimental to the training of machine learning models and have a confounding effect on the assessment of model performance. Nevertheless, employing experts to remove label noise by fully re-annotating large datasets is infeasible in resource-constrained settings, such as healthcare. This work advocates for a data-driven approach to prioritising samples for re-annotation—which we term "active label cleaning". We propose to rank instances according to estimated label correctness and labelling difficulty of each sample, and introduce a simulation framework to evaluate relabelling efficacy. Our experiments on natural images and on a specifically-devised medical imaging benchmark show that cleaning noisy labels mitigates their negative impact on model training, evaluation, and selection. Crucially, the proposed approach enables correcting labels up to 4 × more effectively than typical random selection in realistic conditions, making better use of experts' valuable time for improving dataset quality.

[1] Health Intelligence, Microsoft Research Cambridge, Cambridge CB1 2FB, UK. [2] Department of Radiology, Stanford University, Palo Alto, CA 94304, USA.
[3] These authors contributed equally: Mélanie Bernhardt, Daniel C. Castro. ✉email: ozan.oktay@microsoft.com

The success of supervised machine learning primarily relies on the availability of large datasets with high-quality annotations. However, in practice, labelling processes are prone to errors, almost inevitably leading to noisy datasets—as seen in ML benchmark datasets[1]. Labelling errors can occur due to automated label extraction[2,3], ambiguities in input and output spaces[4], or human errors[5] (e.g. lack of expertise). At training time, incorrect labels hamper the generalisation of predictive models, as labelling errors may be memorised by the model resulting in undesired biases[6,7]. At test time, mislabelled data can have detrimental effects on the validity of model evaluation, potentially leading to incorrect model selection for deployment as the true performance may not be faithfully reflected on noisy data. Label cleaning is therefore crucial to improve both model training and evaluation.

Relabelling a dataset involves a laborious manual reviewing process and in many cases the identification of individual labelling errors can be challenging. It is typically not feasible to review every sample in large datasets. Consider for example the NIH ChestXray dataset[3], containing 112 k chest radiographs depicting various diseases. Diagnostic labels were extracted from the radiology reports via an error-prone automated process[8]. Later, a subset of images (4.5 k) from this dataset were manually selected and their labels were reviewed by expert radiologists in an effort driven by Google Health[2]. Similarly, 30 k randomly selected images from the same dataset were relabelled for the RSNA Kaggle challenge[9]. Such relabelling initiatives are extremely resource-intensive, particularly in the absence of a data-driven prioritisation strategy to help focusing on the subset of the data that most likely contains errors.

Due to the practical constraints on the total number of re-annotations, samples often need to be prioritised to maximise the benefits of relabelling efforts (see Fig. 1), as the difficulty of reviewing labelling errors can vary across samples. Some cases are easy to assess and correct, others may be inherently ambiguous even for expert annotators (Fig. 2). For such difficult cases, several annotations (i.e. expert opinions) may be needed to form a ground-truth consensus[2,10], which comes with increasing relabelling "cost". Hence, there is a need for relabelling strategies that consider both resource constraints and individual sample difficulty—especially in healthcare, where availability of experts is limited and variability of annotations is typically high due to the difficulty of the tasks[11].

While there are learning approaches designed specifically to handle label noise during training, we claim that these strategies can benefit from active labelling for two main reasons: First, clean evaluation labels are often unavailable in practice, in which case one cannot reliably determine whether any trained model is effective for a given real-world application. In that regard, active label collection can iteratively provide useful feedback to NRL approaches. Second, NRL approaches often cope with noise by inferring new labels[12] or disregarding samples[13] that could otherwise be highly informative or even be correctly labelled. However, models trained with these approaches can still learn biases from the noisy data, which may lead them to fail to identify incorrect labels, flag already correct ones, or even introduce additional label noise via self-confirmation. Active label cleaning complements this perspective, aiming to correct potential biases by improving the quality of training dataset and preserving as many samples as possible. This is imperative in safety-critical domains such as healthcare, as model robustness must be validated on clean labels.

Prioritising samples for labelling also underpins the paradigm of active learning, whose goal is to select unlabelled samples that would be most beneficial for training in order to improve the performance of a predictive model on a downstream task. The key difference here for the proposed approach is that our goal is not only to improve model performance but also to maximise the quality of labels given limited resources, which makes it valuable for both training and evaluation of predictive models. In more detail, we demonstrate how active learning and NRL can play complementary roles in coping with label noise.

In this work, we begin by defining the active label cleaning setting in precise terms, along with the proposed relabelling priority score. Using datasets of natural images and of chest radiographs, we then demonstrate experimentally the negative impacts of label noise on training and evaluating predictive models, and how cleaning the noisy labels can mitigate those effects. Third, we show via simulations that the proposed active label cleaning framework can effectively prioritise samples to re-annotate under resource constraints, with substantial savings over naive random selection. Fourth, we analyse how robust-learning[14] and self-supervision[15] techniques can further improve label cleaning performance. Lastly, we validate our choice of scoring function, which accounts for sample difficulty and noise level, comparing with an active learning baseline.

## Results

**Active label cleaning**. In this work, we introduce a sequential label cleaning procedure that maximises the number of corrected

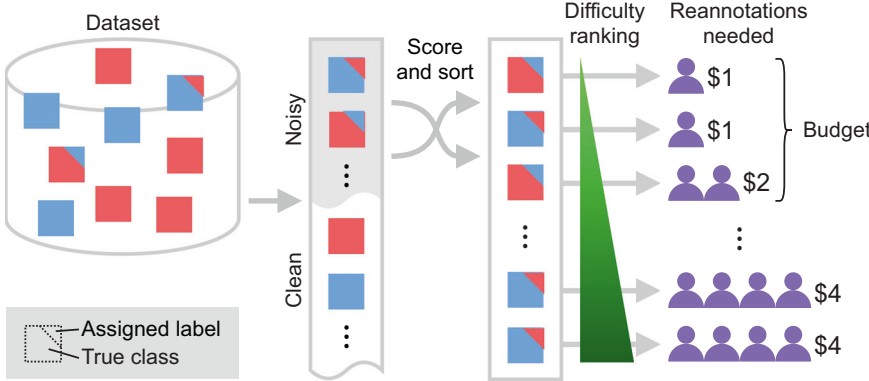

**Fig. 1 Overview of the proposed active label cleaning.** A dataset with noisy labels is sorted to prioritise clearly mislabelled samples, maximising the number of corrected samples given a fixed relabelling budget.

samples under a total resource budget $B \in \mathbb{N}$:

$$\max \quad \underbrace{\frac{1}{N}\sum_{i=1}^{N}\mathbf{1}[\hat{y}_i = y_i]}_{\text{correctness of majority labels}} \quad \text{s.t.} \quad \underbrace{\sum_{i=1}^{N}||\hat{\mathbf{l}}_i||_1 \leq B}_{\text{budget constraint}} \quad ,$$

(1)

where we assume access to a dataset $\mathcal{D} = \{(\mathbf{x}_i, \hat{\mathbf{l}}_i)\}_{i=1}^{N}$, with the $i^{\text{th}}$ image $\mathbf{x}_i$, label counts vector $\hat{\mathbf{l}}_i \in \mathbb{N}^C$ with $C$ classes, and corresponding majority label $\hat{y}_i = \mathrm{argmax}_{c \in \{1,\dots,C\}} \hat{l}_i^c$. All samples are assumed to initially contain at least one label ($||\hat{\mathbf{l}}_i||_1 = \sum_{c=1}^{C} \hat{l}_i^c \geq 1$), and some instances are mislabelled, i.e. their initial (majority) label $\hat{y}$ deviates from the true class $y$. Unlike the traditional active learning objective, we aim to achieve a clean set of labels that can later be used not only for model training but also for benchmarking purposes. In that regard, our setting differs from active learning by allowing for re-annotation of already labelled instances as in 're-active' learning[16]. This is an important consideration as there is often a trade-off between label quality and downstream model performance[16,17]. Yet, conventional approaches[18,19] may not be practically feasible with modern deep learning, as they often entail repeated retraining of models to measure the impact of each sample.

The proposed framework (see Box Item Algorithm 1) determines relabelling priority based on predicted posteriors from a trained classification model, $p_{\theta}(\hat{y}|\mathbf{x})$, parametrised by $\theta$. Label cleaning is performed over multiple iterations and at each iteration either a single or a batch of samples are relabelled. Within an iteration, samples are first ranked according to predicted label correctness and ambiguity (i.e. annotation difficulty). Then, each prioritised sample is reviewed sequentially by different annotators until a majority is formed among all the collected labels $\hat{\mathbf{l}}_i$ (i.e. $\hat{l}_i^{\hat{y}_i} > \hat{l}_i^c, \forall c \neq \hat{y}_i$, with $\sum_{c=1}^{C} \hat{l}_i^c > 1$). In the next iteration, the remaining samples are re-prioritised and the process repeats until the relabelling budget ($B$) is exhausted. Assuming that each annotation has a fixed cost, the more annotations a sample requires until a majority is reached, the more expensive its relabelling. Hence, to achieve the objective in Eq. (1), clearly mislabelled samples need to be prioritised over difficult cases, and correctly labelled samples should have the lowest relabelling priority.

To this end, we propose to rank available samples by the following scoring function $\Phi$:

$$\Phi(\mathbf{x}, \hat{\mathbf{l}}; \theta) = \underbrace{\mathrm{CE}(\hat{\mathbf{l}}, p_{\theta})}_{\text{noisiness} \uparrow} - \underbrace{\mathrm{H}(p_{\theta})}_{\text{ambiguity} \downarrow} .$$

(2)

The first term, defined as the cross-entropy from the normalised label counts to the predicted posteriors,

$$\mathrm{CE}(\hat{\mathbf{l}}, p_{\theta}) = -\mathbb{E}_{\hat{\mathbf{l}}/||\hat{\mathbf{l}}||_1}\left[\log p_{\theta}(\hat{y}|\mathbf{x})\right],$$

(3)

corresponds to the estimated noisiness (i.e. negative log-likelihood) of the given labels. Cross-entropy has been commonly used in NRL to detect mislabelled samples by ranking w.r.t. loss values[14,20,21], thus aiming to maximise the objective in Eq. (1).

On the other hand, obtaining a majority label vote requires different numbers of re-annotations for different images, depending on their difficulty. This is quantified by the entropy term $\mathrm{H}(p_{\theta})$ defined over posteriors,

$$\mathrm{H}(p_{\theta}) = -\mathbb{E}_{p_{\theta}(\hat{y}|\mathbf{x})}\left[\log p_{\theta}(\hat{y}|\mathbf{x})\right],$$

(4)

which penalises ambiguous cases in the ranking. This same quantity is employed as an estimate of aleatoric uncertainty (i.e. data ambiguity) in active learning methods[22,23], which deprioritises ambiguous samples to address annotation budget constraints like ours in Eq. (1). This quantity can be better estimated by marginalising it over a distribution of $\theta$ (Supplementary Fig. 6 and Table 2). Similar objectives have also been used in the contexts of semi-supervised learning[24] and entropy regularisation[25]. In Methods section, we present different options for the predictive model $p_{\theta}$ to be used in computing these quantities.

Lastly, the proposed sequential framework allows for regular model updates during the cleaning process; as such, the selector model can be fine-tuned at any time using the corrected labels collected so far (see Box Item Algorithm 1 Line 13)—which has shown to improve cleaning performance (Supplementary Table 3).

**Datasets used in the experiments.** To analyse label noise scenarios, we experiment with two imaging datasets, namely CIFAR10H and NoisyCXR, containing multiple annotations per data point, which helps us to model the true label distributions and associated labelling cost. In CIFAR10H[5,26], on average 51 manual annotations were collected for each image in the test set

---

**Algorithm 1. Active label cleaning**

```
Given:   Y = {lᵢ}ᵢ₌₁ᴺ: True label distributions
Input:   𝒟 = {(xᵢ, l̂ᵢ)}ᵢ₌₁ᴺ: Dataset with noisy labels
         B ∈ ℕ: Relabelling budget
         b ∈ ℕ: Update frequency

 1:  θ ← TRAINROBUSTMODEL(𝒟)
 2:  ℐ_avail ← {1,...,N},   ℐ_cleaned ← ∅
 3:  count ← 0
 4:  while count < B do                                    ▷ If budget remains
 5:      j ← arg max_{i∈ℐ_avail} Φ(xᵢ, l̂ᵢ; θ)              ▷ Rank (Eq. (2))
 6:      repeat
 7:          l̂ⱼ ← l̂ⱼ + SAMPLE(lⱼ)                          ▷ Acquire one-hot label
 8:          count ← count + 1
 9:      until majority formed in l̂ⱼ
10:      ℐ_avail ← ℐ_avail \ {j},   ℐ_cleaned ← ℐ_cleaned ∪ {j}
11:      𝒟 ← {(xᵢ, l̂ᵢ) : i ∈ ℐ_avail ∪ ℐ_cleaned}
12:      if count divisible by b then
13:          θ ← UPDATE(θ, 𝒟)                              ▷ Fine-tune model
14:      end if
15:  end while
16:  return 𝒟
```

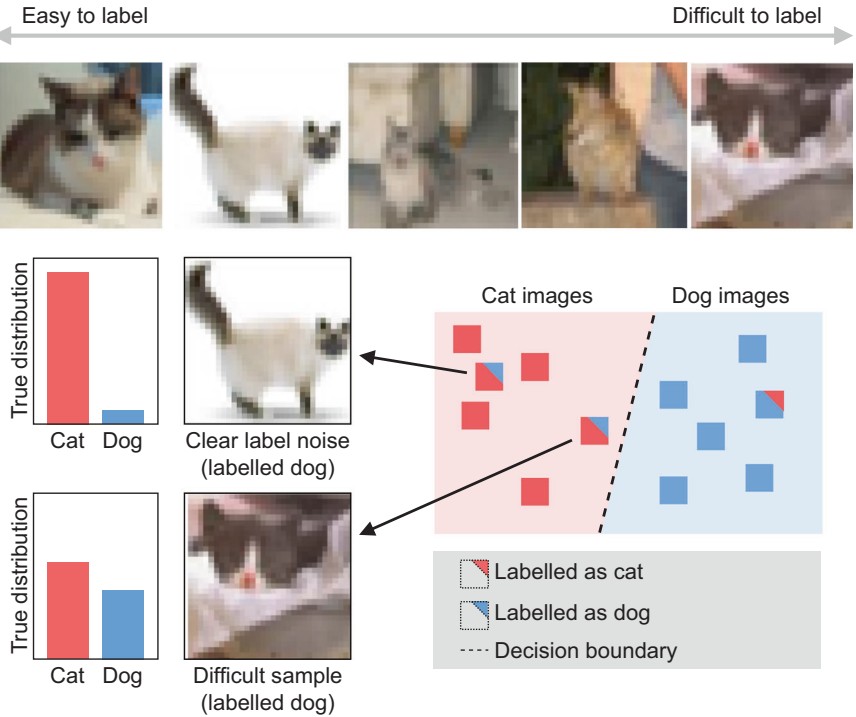

**Fig. 2 Image labelling can become difficult due to ambiguity in input space[26].** Top row shows the spectrum of ambiguity for cat images sampled from CIFAR10H dataset. The 2D plot illustrates different types of mislabelled samples: clear noise and difficult cases. We expect the former to be adjacent to semantically similar samples with a different label, and the latter to be closer to the optimal decision boundary.

of CIFAR10[27], a collection of 10 k natural images grouped in 10 classes. The experiments on CIFAR10H are intended to understand the challenges associated with different noise rates, noise models, and active relabelling scenarios. We also set up a benchmark for label cleaning on medical images, NoisyCXR, comprising 26.6 k chest radiographs and multiple labels from clinical datasets indicating the presence of pneumonia-like opacities[3,9]. This serves as a more challenging imaging benchmark, where modelling difficulties are coupled with the challenges associated with noisy labels and sample prioritisation.

**Label noise undermines both model training and evaluation.** In particular, training with noisy labels is known to impair the performance of predictive models[6,7], as the latter may be forced to memorise wrong labels instead of learning generalisable patterns. As an example, a vanilla convolutional neural network (CNN) classifier trained on CIFAR10H with clean labels achieves 73.6% accuracy on a clean test set of 50 k images. However, if the same model is trained on a version of this dataset with 30% label noise, its test accuracy degrades to 64.1%, a substantial drop of 8.5%. Even a noise-robust classifier, initialised with self-supervision (i.e. pre-trained with images only), drops from 80.7% to 78.7% in accuracy when fine-tuned on the same noisy training labels as opposed to clean labels.

We also highlight the adverse implications of validation label noise on model evaluation. Its most evident impact is that true model performance can be underestimated. For instance, consider CIFAR10H with 40% noise rate in both training and validation labels (instance-dependent noise[28]; see Methods). A self-supervised classifier, as above, is estimated to be only 46.5% accurate on these noisy validation labels, compared to 65.8% on the true, clean labels. Additionally, model rankings based on noisy validation metrics may be unreliable, leading to misinformed model selection or design decisions. For example, a

second classifier, trained on the same training set via co-teaching[14], achieves a lower 45.9% accuracy on the same noisy validation set, while its real accuracy (unobservable in practice) on clean labels is 69.9%. In other words, the worst performing model (here, self-supervised) would have been chosen for deployment because of misleading validation metrics obtained on noisy labels.

In summary, label noise can negatively affect not only model building, but also validation. The latter is especially relevant in high-risk applications, e.g. for the regulation of models in healthcare settings. Our results in later sections demonstrate how active cleaning of noisy training and evaluation labels can help mitigate such issues.

**Simulation of sequential relabelling.** A traditional way to evaluate a label cleaning algorithm is via its ability to separate noisy from clean labels (e.g. detection rate)[18,19]. However, such an evaluation does not necessarily take into account the sequential nature of sample selection, label acquisition, and model updates. Further, it typically assumes all new labels are correct, neglecting the effect of sample ambiguity on how many annotations are required. Both are crucial factors in measuring the effectiveness of label cleaning efforts in terms of resource usage and final outcome.

To account for these factors, we propose a realistic simulation of the entire relabelling process that leverages the true label distribution of each sample. As shown in Box Item Algorithm 1, the simulation proceeds sequentially. Once a sample has been selected for relabelling, new labels are drawn from its true distribution (Box Item Algorithm 1, Line 7), which reflects the probabilities of real annotators assigning each class to a given sample. This enables more faithful modelling of the expected relabelling effort for each sample and of the likely label errors. Through this simulation, the performance of sample selection

**Table 1 Classification accuracy (%) before and after label cleaning.**

|  | Selector | Scoring | Classifier | Before cleaning | After cleaning |
|---|---|---|---|---|---|
| (1) | Vanilla | Eq. (2) | Vanilla | 64.1 | 68.4 |
| (2) | Vanilla | BALD[34] | Vanilla | 64.1 | 68.3 |
| (3) | SSL | Eq. (2) | Vanilla | 64.1 | 70.9 |
| (4) | SSL | Eq. (2) | SSL | 78.7 | 80.3 |
| (5) | Vanilla | Eq. (2) | SSL | 78.7 | 79.4 |
| (6) | Co-teaching | Eq. (2) | Co-teaching | 66.5 | 68.8 |
| (7) | – | – | ELR[29] | 67.0 | – |
| (8) | (Clean training) |  | Vanilla | 73.6 | – |
| (9) | (Clean training) |  | SSL | 80.7 | – |

Models are evaluated on a clean test set ($N = 50$ k) before and after relabelling 32.7% of samples in the training set (CIFAR10H, $N = 5$ k, $\eta = 30\%$).

algorithms can be measured in terms of percentage of corrected labels in the dataset as a function of the number of re-annotations.

The relabelling simulation is used as a testbed to evaluate three different sample ranking algorithms: standard CNN (vanilla), co-teaching[14], and self-supervised pre-training with fine-tuning[15]. All approaches employed the same image encoder (ResNet-50), augmentations, and optimiser type (see supplement for implementation details). At each iteration, new labels are sampled from the true label distribution, and experiments are repeated with 5 independent random seeds. Performance of selection algorithms is compared to two baselines: the oracle and random selector, which determine respectively the upper and lower bounds. The random selector chooses the next sample to be annotated uniformly at random as done in previous studies[2,9]. Conversely, the oracle simulates the ideal ranking method by having access to the true label distribution. This enables the oracle to prioritise least difficult noisy samples first, hence maximising the number of corrected samples for a given resource constraint. Because the oracle may still sample incorrect labels, we additionally include a "minimal sampler" for reference, which systematically returns the ground-truth label without emulating the ambiguity of each sample. The maximum relabelling budget ($B$) in the simulation is set to the expected number of re-annotations required by the oracle to clean all mislabelled samples.

**Cleaning noisy training labels improves predictive accuracy**. All label cleaning models are first trained on their respective training datasets (CIFAR10H and NoisyCXR), and subsequently used to rank samples within the same set for relabelling. In the rest of the manuscript, the term "selector" refers to the models used for label cleaning. Table 1 shows that all training methods benefit from newly acquired labels regardless of their underlying weight initialisation (e.g. self-supervised learning—SSL) and noise-robust learning (NRL) strategy, as in ELR[29]. In that regard, label cleaning serves as a complementary approach to NRL by recycling data samples containing noisy labels. Additionally, the comparison between rows 3 and 7 show that acquiring a new set of labels can yield significantly better results than state-of-the-art NRL models trained on noisy labels, which performs closer to the upper-bound (row 8) where training is done with all true labels. Class imbalance, inherent noise model, and difficulty of samples are expected to introduce challenges to most NRL methods.

**Label cleaning can be done in a resource-effective manner**. In Table 1, we also observe that sample ranking methods (selectors) can influence the outcome of label cleaning efforts in terms of predictive accuracy (see rows 1 vs 3 and rows 4 vs 5). To further investigate the impact of selector types, we carried out label cleaning simulations (see Fig. 3) on both CIFAR10H and

NoisyCXR datasets for initial noise rates of $\eta = 15\%$ and 12.7%, respectively. The methods are compared quantitatively regarding fraction of corrected labels (i.e. reduction in noise rate) for varying number of re-annotations. Additionally, the sample ranking is visually illustrated in Fig. 4 along with the predicted label correctness and ambiguity scores.

It is apparent that, for a fixed improvement in number of corrected labels, all proposed selectors require much fewer re-annotations than random. For instance, the vanilla selector can reach 90% correct labels with $3 \times$ (NoisyCXR) or $2.5 \times$ (CIFAR10H) fewer re-annotations than random. More specifically, we can see that NRL (here, co-teaching) benefits selection algorithms in prioritising noisy labels. Additionally, SSL yields further performance improvements by learning generic semantic representations without requiring labels, which in principle reduces chances of memorising noise. In that regard, it is a complementary feature to traditional NRL approaches as long as there is enough data for training. In particular, SSL pre-training yields improved noisy sample detection on NoisyCXR dataset in comparison to initialisation from scratch or pre-trained ImageNet weights. Figure 5 exemplifies clear noisy and difficult cases for NoisyCXR. We observed that the entropy term in Eq. (2) has had a smaller effect on the cleaning performance with NoisyCXR—which we hypothesise may be related to its binary labels—thus was not included in the simulation in Fig. 3a.

The experiments are also repeated for larger noise rates (see Supplementary Fig. 2) to identify the limits beyond which sample selection algorithms deteriorate towards random sampling. The results show similar performance as long as diagonal dominance[30] holds on average in class confusion matrices (see Fig. 6), which is verified by experimenting with uniform and class-dependent noise models[31].

**Sample scoring function impacts cleaning performance**. We study the influence of the self-entropy term ($H(p_\theta)$) used in Eq. (2) in an ablation study. We see that selectors prioritising clear label noise cases yield higher numbers of corrected label noise, and this effect is further pronounced by including the entropy term in the scoring function (see Supplementary Fig. 3).

Additionally, ranking samples for labelling is related to active learning, whereby one annotates unlabelled instances in order to maximise model performance metrics. Typical active-learning methods attempt to identify the most informative examples[32], relying on e.g. core-sets[33] or high predictive uncertainty[22]. A representative example is Bayesian active learning by disagreement (BALD)[22,34], which quantifies the mutual information between each sample's unknown label and the model parameters. However, these criteria often prioritise under-represented parts of the data space, in particular samples close to the decision boundary. This may not necessarily coincide with samples with

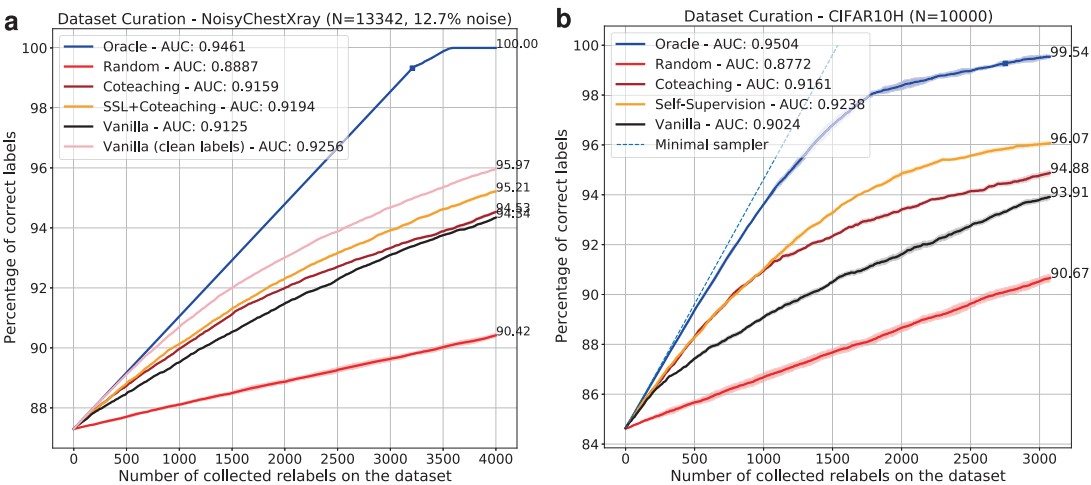

**Fig. 3 Results of the label cleaning simulation on training datasets.** **a** NoisyCXR ($\eta = 12.7\%$); **b** CIFAR10H ($\eta = 15\%$). For a given number of collected labels (x-axis), a cost-efficient algorithm should maximise the number of samples that are now correctly labelled (y-axis). The correctness of acquired labels is measured in terms of accuracy. The area-under-the-curve (AUC) is reported as a summary of cleaning efficiency of each selector across different relabelling budgets. The upper and lower bounds are set by oracle (blue) and random sampling (red) strategies. The pink curve (**a**) illustrates the practical "model upper bound" of cleaning performance when the selector model is trained solely on clean labels, its performance being bound to the capacity of the model to fit the data. Shaded areas represent ± standard deviation over 5 random seeds for relabelling.

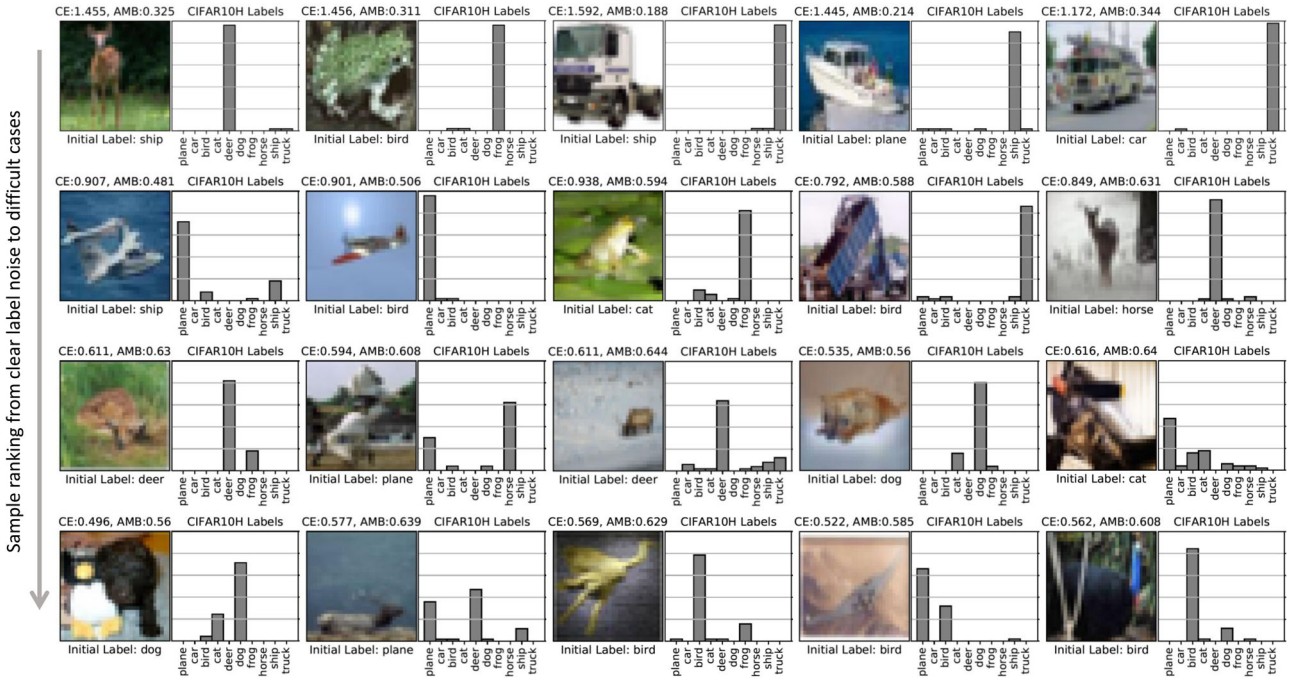

**Fig. 4 Ranking of CIFAR10H samples (15% initial noise rate) by the SSL-Linear algorithm.** The top row illustrates a representative subset of images ranked at the top-10 percentile with the highest priority for relabelling. Similarly, the second and third rows correspond to 25–50 and 50–75 percentiles, respectively. At the bottom, ambiguous examples that fall into the bottom 10% of the list ($N = 2241$) are shown. Each example is shown together with its true label distribution to highlight the associated labelling difficulty. This can be compared against the label noisiness (cross-entropy; CE) and sample ambiguity (entropy; AMB) scores predicted by the algorithm (see Eq. (2)), shown above each image. As pointed out earlier, adjudication of samples provided at the bottom does require a large number of re-annotations to form a consensus. The authors in ref. [26] explore the causes of ambiguity observed in these samples.

noisy labels, in particular clear noise cases (see Fig. 2). To analyse the suitability for label cleaning of such an approach, we repeat these experiments with the BALD score in replacement of the function in Eq. (2), using the same relabelling budget. Rows 1–2 in Table 1 show that classification accuracy gains on the test set are comparable between the two. However, the label quality improvement is significantly inferior for BALD (Supplementary Fig. 3), as it favours samples with the highest disagreement—which may not correspond to label noise. Hence, whereas both scoring functions (BALD and Eq. (2)) yield similar model performance improvements, Eq. (2) offers the advantage of improving the quality of noisy datasets.

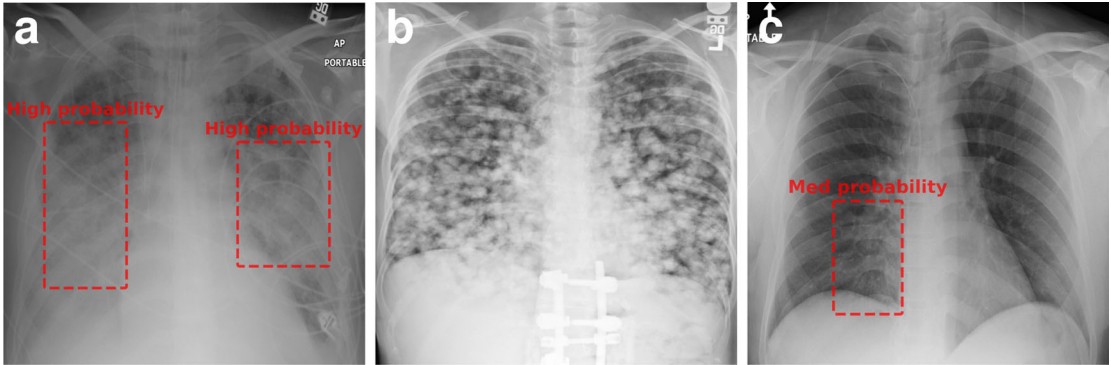

**Fig. 5 Chest X-ray images selected from NoisyCXR dataset, which do not contain "pneumonia" label in the NIH dataset[3]. a** Correctly identified noise case with pneumonia-like opacities shown with bounding boxes. **b** Wrongly flagged sample with a correct label; here the model confuses lung nodules with pneumonia-like opacities. **c** A difficult case with subtle abnormality where radiologists indicated medium-confidence in their diagnosis as shown by the highlighted region (RSNA study[9]).

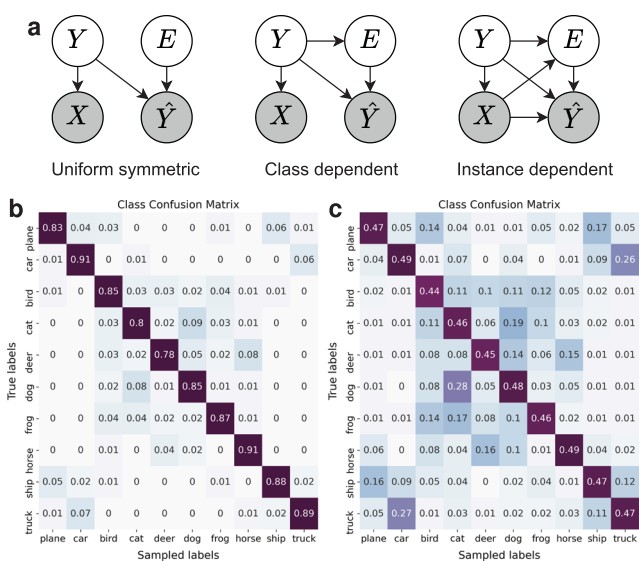

**Fig. 6 Understanding label noise patterns. a** Different label noise models used in robust learning. The statistical dependence between input image ($X$), true label ($Y$), observed label ($\hat{Y}$), and error occurrence ($E$) is shown with arrows (adapted from Frénay et al.[55]). **b–c** CIFAR10H class confusion matrices (temperature $\tau = 2$) for all samples (**b**) and difficult samples only (**c**).

**Cleaning evaluation labels mitigates confounding in model selection.** In NRL[12,14,21,35], validation data is traditionally assumed to be perfectly labelled and can be relied on for model benchmarking and hyperparameter tuning. Furthermore, classification accuracy has been shown to be robust against noise[30,36] assuming that labelling errors and inputs are conditionally independent, $p(\hat{y}|y, \mathbf{x}) = p(\hat{y}|y)$. While these assumptions may be valid in certain applications, they often do not hold for real-world datasets, wherein the labelling process clearly relies on examining the inputs. Thus, models can learn biases from the training data that would correlate with labelling errors in the evaluation set (see Fig. 6a).

To illustrate the pitfalls of evaluating models on noisy labels, we have conducted experiments on CIFAR10H using symmetric (SYM) and instance-dependent noise (IDN) models[28] (see supplement for details). In these experiments, diagonal dominance[30] is preserved for all confusion matrices, and the statistical dependence between labelling errors, input, and true label are the same in both training and validation sets. Furthermore, all the available relabelling resources are used for cleaning the noisy labels in the evaluation set instead of training data to obtain an unbiased estimate of model ranking and performance. Though resources can

be split to partially clean both training and evaluation sets without changes to the algorithm, we generally recommend prioritising the cleaning of evaluation data up to an acceptable level before applying any remaining budget to clean the training set.

Here, ResNet-50 models[37] with different regularisation settings are trained with noisy training labels ($|\mathcal{D}_{\text{train}}| = 10$ k) and are later used to clean labels on evaluation set ($|\mathcal{D}_{\text{eval}}| = 50$ k). Labels for sets $\mathcal{D}_{\text{train}}$ and $\mathcal{D}_{\text{eval}}$ are determined for each experiment separately using SYM and IDN noise models with 40% noise rate on average. The experimental results (Table 2) show that standard performance metrics and model ranking strongly depend on the underlying conditions leading to labelling errors, which shows the need for clean validation sets in practical applications for model selection.

To tackle these challenges, we applied the active relabelling procedure (SSL-Linear) on this noisy validation set by using the same corresponding training data ($\mathcal{D}_{\text{train}}$). Through sample prioritisation and relabelling of 10% of the entire set ($\mathcal{D}_{\text{eval}}$), the bias in model selection can be alleviated as shown in Table 2. At the end of the label cleaning procedure, the noise rates are reduced to 31.32% (IDN) and 30.25% (SYM); in other words, 86.8% (IDN) and 97.5% (SYM) of the selected images were

**Table 2 Classification accuracy under noisy evaluation labels.**

| Model | True | Noisy (40%) | Cleaned (10%) | Cleaned (20%) |
|---|---|---|---|---|
| SYM | | | | |
| M1 | 73.32 | 45.19 (0.21) | 54.37 (0.13) | 62.53 (0.09) |
| M2 | **80.16** | **48.93** (0.23) | **58.45** (0.10) | **67.42 (0.10)** |
| IDN | | | | |
| M1 | **69.91** | 45.93 (0.10) | 49.97 (0.06) | **55.87** (0.06) |
| M2 | 65.76 | **46.50** (0.13) | 49.90 (0.12) | 55.28 (0.10) |

Co-teaching (M1) and SSL (M2) models are compared on a noisy CIFAR10H validation set $\mathcal{D}_{eval}$ over three runs using different label initialisations. Both approaches use ResNet-50 with different weight initialisation and regularisation. We compare classification accuracy on true, noisy, and cleaned labels. M2 is deliberately less regularised (i.e. weight decay) and is expected to perform worse on a more challenging IDN noise model. For each validation set, the highest accuracy is highlighted in bold.

initially mislabelled and then corrected. Here, IDN model is observed to be more challenging for identifying noisy labels.

## Discussion

This work investigated the impact of label noise on model training and evaluation procedures by assessing its impact in terms of (I) predictive performance drop, (II) model evaluation results, and (III) model selection choices. As potential ways to mitigate this problem can be resource-demanding depending on the application area, we defined cost-effective relabelling strategies to improve the quality of datasets with noisy class labels. These solutions are benchmarked in a specifically-devised simulation framework to quantify potential resource savings and improvement in downstream use-cases.

In particular, we highlight the importance of cleaning labels in a noisy evaluation set. We showed that neglecting this step may yield misleading performance metrics and model rankings that do not generalise to the test environment. This can, in turn, lead to overoptimistic design decisions with negative consequences in high-stakes applications such as healthcare. One of our main findings is that the patterns of label error in the data (i.e., structural assumptions about label errors shown in Fig. 6) can have as large an impact on the efficacy of label cleaning and robust-learning methods as the average noise rates, as evidenced by the results obtained on both training and validation sets. We therefore recommend carefully considering the underlying mechanisms of label noise when attempting to compare possible solutions.

Note that, when cleaning the test set, there may be concerns about introducing dependency between the training and test sets. To avoid this, selector models utilised for sample prioritisation should ideally (i) be trained solely on the test set and (ii) not be used for classification and evaluation purposes. Moreover, it is worth noting that modelling biases could be reflected in the ranking of the samples. Such biases may be mitigated by employing an ensemble of models with different formulations and inductive biases for posterior estimation in Eq. (2), as the framework makes no assumptions about the family of functions that can be used for label cleaning.

The results also suggest that even robust-learning approaches may not fully recover predictive performance under high noise rates. In such cases, SSL pre-training is experimentally shown to be a reliable alternative, outperforming noise-robust models trained from scratch, even more so with the increasing availability of unlabelled datasets. Lastly, we show that acquiring new labels can complement NRL by recycling data samples even if their labels are noisy, and can also handle biased labels. Thus, the two domains can be combined to obtain not only a better model, but also clean data labels for downstream applications.

A limitation of data-driven approaches for handling label noise is that they may still be able to learn from noise patterns in the data when label errors occur in a consistent manner. As such, some mislabellings may remain undetected. However, it is worth noting that our approach does not flip already correct labels (assuming that manual labellers provide i.i.d. samples from the true data distribution). From this perspective, the proposed algorithm will converge towards the true label distribution (with a sufficient number of labels), addressing label inconsistencies detectable by the selector model while optimising for the objective given in Eq. (1). Under extreme conditions where the bounded noise rate assumption may not hold (i.e. where on average there are more incorrect labels than correct in a given dataset), random selection can become preferable over data-driven approaches[38]. However, in the case of bounded noise rate—as in most real-world applications—the active learning component of the proposed framework can potentially address such consistent noise patterns in labels[38]. Indeed, the proposed active approach enables establishing a distilled set of expert labels to tackle this challenge, instead of solely relying on self-distillation or hallucination of the true labels as in NRL methods[38]. To further extend our methodology along these lines, one could treat the newly acquired labels as expert-distilled samples, and rely more heavily on them for posterior updates in the proposed iterative framework. Lastly, recall that the proposed label cleaning procedure is a human-in-the-loop system. Therefore, from a practical point-of-view, the process can be monitored and intervened upon whenever assumptions may be violated or if there is a concern around mislabelling biases in the dataset.

Although the present study focused on imaging, the proposed methodology is not limited to this data modality, and empirical validation with other input types is left for future work. It will also be valuable to explore, for example, having the option to also annotate unlabelled samples, or actively choosing the next annotator to label a selected instance. Such extensions to active cleaning will significantly broaden its application scope, enabling more reliable deployment of machine learning systems in resource-constrained settings.

Here, we assume that the majority vote was representative of the ground truth. However, in some cases majority vote can become suboptimal when annotators have different levels of experience in labelling data samples[10]. For these circumstances, future work could explore sample selection objectives and label assignment taking into account the expertise of each annotator. Similarly, multi-label fusion techniques[39–42] can be used within the proposed label cleaning procedure to restore true label distribution by modelling labelling process and aggregating multiple noisy annotations. Such approaches critically rely on the availability of multiple labels for each sample—which can be realised towards the end of relabelling efforts.

## Methods

**CIFAR10H dataset**. CIFAR10H[5,26] is an image dataset comprising the test set of CIFAR10 (10 k images) and multiple labels collected from crowdsourcing. Each image on average contains 51.1 labels distributed over 10 class categories. In our study, the true label distributions were calculated by normalising the histogram of these labels on a per image basis. In the experiments, we assume that our initial dataset contains labelling errors. These partially noisy labels were obtained by sampling the initial label by taking into account the label distribution for each sample. The noise rate in experiments is controlled by applying temperature scaling[43] ($\tau$) to each distribution, which preserves the statistical dependence between input images and their corresponding labels. The higher the temperature, the closer to a uniform distribution the label distribution becomes, hence the higher the probability of sampling a noisy label for a given sample.

Samples in this dataset are not all equally difficult to label or classify[26]. For instance, for some images, all annotators agreed on the same label, for some others collected labels were more diverse as the image was harder to identify. Hence, we defined sample difficulty as a function of the normalised Shannon entropy of its

**Table 3 Correspondence of chest radiography labels from the RSNA challenge[9], NIH[3], and NoisyCXR datasets.**

| | RSNA labels[9] | |
| --- | --- | --- |
| | Pneumonia-like opacity | No pneumonia-like opacity |
| NIH labels[3]: | | |
| Pneumonia | 367 | **441** |
| Consolidation/infiltration (not pneumonia) | *3,988* | *8551* |
| Other diseases only | **1101** | 5567 |
| No finding | **556** | 6113 |
| NoisyCXR labels: | | |
| Pneumonia-like opacity | 3956 | **1296** |
| No pneumonia-like opacity | **2056** | 19,376 |
| Total | 6012 | 20,672 |

The disagreements between two label sets are highlighted in bold font. Fields in italic indicate an unclear agreement between both sets of labels. NoisyCXR labels are collected from the original NIH labels where possible, and the remainder are uniformly sampled (10%) from the "Consolidation/infiltration" category to increase the noise rate further in the experiments.

target distribution, $\sum_{c=1}^{C} p_c \log_C p_c$. If the entropy associated to this distribution was higher than a certain threshold (0.3 in our experiments), the sample was classified as "difficult". The entropy of the target distribution can also be related to the expected number of relabellings required for this sample (see problem definition in Results). In Fig. 6b, we show the average class confusion matrices over the CIFAR10H dataset for $\tau = 2$. In Fig. 6c, the confusion matrix of only the difficult samples is shown to highlight the classes that are confused the most in a standard labelling process.

**NoisyCXR: a medical benchmark dataset for label cleaning.** In the experiments, we used a subset of the medical image dataset released by the NIH[3]. The original dataset contains 112 k chest radiographs with labels automatically extracted from medical records using a natural language processing algorithm. In the original data release, labels are grouped into 14 classes (non-mutually exclusive) indicating the presence of various diseases or no finding at all. However, there have been studies[8] showing errors in these auto-extracted image labels. Later, the RSNA released the Pneumonia Detection Challenge[9] aiming to detect lung opacities indicative of pneumonia. The challenge dataset is comprised of 30 k images randomly sampled from the NIH ChestXray dataset mentioned above. To adjudicate the original labels released by the NIH, a board of radiologists reviewed and re-assessed each image on this dataset and delimited the presence of lung opacities associated to pneumonia. Samples are hence classified into two categories: "pneumonia-like lung opacity" and "no pneumonia-like lung opacity". Here we aim to utilise a realistically noisy version of this RSNA dataset encountered in real-world application scenarios. To this end, the original NIH labels are leveraged to construct a noisy version of the true labels released by the RSNA. In more detail, we analysed the data in four categories (see rows in Table 3) based on their original NIH labels and the class taxonomy[44] in order to map this original label to a binary label indicating presence or absence of pneumonia-like opacities:

- Cases where the NIH label contains "Pneumonia". All these samples were assigned to the "pneumonia-like opacity" category in NoisyCXR.
- Cases associated to opacities potentially linked to pneumonia: NIH label contains "consolidation" or "infiltration" (but not "pneumonia"). For these cases, it is unclear whether the original NIH label should be mapped to "pneumonia-like opacity" or not[44,45]. Hence, for these cases, in NoisyCXR, we attributed the correct (final RSNA) label to 90% of the cases and flipped the label of the remaining 10%.
- Cases tagged as "no finding", all assigned to the "no pneumonia-like opacity" category.
- Cases linked to other pathology (i.e., original label only contains pathology unrelated to pneumonia e.g. pleural effusion, nodules, fluid[3]), all assigned to the "no pneumonia-like opacity" category.

The obtained binary mapping hence gives us a set of noisy labels for the "pneumonia-like opacity" classification task whereas the labels released as part of the challenge are considered as the correct labels for the data cleaning experiment. The final noisy dataset contains 26,684 images with a noise rate of 12.6% (see Table 3). As in the case of CIFAR10H, the uncertainty in the final labels permits to separate ambiguous from easy cases. In particular, using the confidence associated to each pneumonia-like lung opacity bounding box[46] we distinguished cases as follows:

- If the final label was "Pneumonia-like opacity" but all bounding boxes were of low confidence, we considered the case as difficult and define the label

distribution for relabel sampling to $\mathbb{P}_{label}($ Pneumonia opacity $) = 0.66$. If at least one of the boxes had a medium or high confidence score, the case is considered easy and we set $\mathbb{P}_{label}($ Pneumonia opacity $) = 1$.
- If the final label was "No Pneumonia-like opacity" but some readers delineated opacities, the sample is considered ambiguous and it's class label is set to $\mathbb{P}_{label}($ Pneumonia opacity $) = 0.33$, Otherwise, it is considered easy and $\mathbb{P}_{label}($ Pneumonia opacity $) = 0$.

**Learning noise-robust image representations.** NRL is essential for handling noise and inconsistencies in labels which can severely impact the predictive performance of models[6,7]. Although, this type of machine learning is most commonly employed to improve model's predictive performance, in our setting, we use NRL methods to obtain an unbiased sample scoring function in active label cleaning. Interested readers should refer to the extended literature review[47] for further information on this topic.

The sample selection algorithms presented in the next section heavily rely on models trained with NRL methods; in particular, we are utilising methods from two broad categories: model regularisation[48–50] and sample exclusion approaches[13,14,20]. The former set of methods have been shown to be effective against memorising noisy labels[6], by favouring simpler decision boundaries. Sample exclusion approaches, on the other hand, aim to identify noisy samples during training by tracking loss values of individual samples[13,14] or gradient vector distributions[29,51]. Alternative meta-model based approaches are not considered in this study as unbiased clean validation set may not always be available in real-world scenarios.

**Sample selection algorithms.** Here we propose three selection algorithms for accurately estimating class posteriors, $p_\theta(\hat{y}|\mathbf{x})$, to prioritise mislabelled data points in the scoring function $\Phi$ (see Eq. (2)). In detail, we first train deep neural networks with noise robustness, then use them to identify the corrupted labels. The methods differ in how the robustness is introduced and how the networks are used afterwards. These methods were proposed and studied to take into account different noise cleaning setups, including the number of labelled and unlabelled samples, and prior knowledge on the expected noise rate.

As a baseline approach, networks are trained with noisy labels and augmented images by minimising a negative log-likelihood loss term, which is referred to as vanilla. This approach is expected to be biased by the noisy labels and to perform sub-optimally in prioritising samples for relabelling.

*Supervised co-teaching training.* To cope with noisy labels, we first propose to use co-teaching[14] in a supervised learning setting, which has shown promising performance in classification accuracy and robustness. In the co-teaching scheme, two identical networks $f_{\theta_1}$ and $f_{\theta_2}$ with different initialisation are trained simultaneously, but the batch of images at each training step for $f_{\theta_1}$ is selected by $f_{\theta_2}$ and vice versa. The rationale is that images with high loss values during early training are ones that are difficult to learn, because their labels disagree with what the network has learned from the easy cases—indicating these labels might be corrupted. Also, using two networks rather than one prevents a self-confirmation loop as in[52]. Co-teaching thus provides a method for robust training with noisy labels that can be employed to identify corrupted labels. We ensemble the predictions of both trained networks as $\frac{1}{2}\sum_m f_{\theta_m}(\mathbf{x}_i) = \hat{\pi}_i$, where $\hat{\pi}_i \in [0,1]^C$ and $\sum_{c=1}^{C} \hat{\pi}_i^c = 1$, and use the predicted distribution, $\hat{\pi}_i$, in the scoring function $\Phi$ to rank the samples.

*Self-supervision for robust learning.* NRL can still be influenced by label noise and exclude difficult samples, as they yield large loss values. SSL is proposed as a second approach to address this and further improve the sample ranking performance whenever an auxiliary task can be defined to pre-train models. As the approach does not require target labels and only extracts information from the images, it is an ideal way to learn domain-specific image encoders that are unbiased by the label noise. It can also be used in conjunction with the co-teaching algorithm in an end-to-end manner.

To this end we use BYOL[15] to learn an embedding for the images, i.e., a non-linear transformation $\mathbf{x}_i \mapsto g_\theta(\mathbf{x}_i) = \mathbf{h}_i$, where $\mathbf{h}_i \in \mathbb{R}^L$ and $L$ is the size of the embedding space. As SSL is agnostic to the labels, the embedding will be formed based only on the image content, such that similar images will be close in the embedding space and dissimilar images will be further apart. The learnt encoder can later be used to build noise-robust models. One way to achieve this is by fixing the weights of $g_\theta$ and train a linear classifier on top of the learned embedding as $\mathbf{h}_i \mapsto f_\varphi(\mathbf{h}_i) = \hat{\pi}_i$, which is referred to as "SSL-Linear". To obtain better calibrated posteriors, large logits are penalised at training time with function $\gamma(x) = \alpha \cdot \tanh(x/\alpha)$[15] and optimised with smoothed labels[53]. The linear classifier can only introduce a linear separation of the images according to their given labels. Assuming images with similar contents (i.e. nearby in embedding space) should have similar labels, the simple linear decision boundary introduces robustness to mislabels.

**Reporting summary**. Further information on research design is available in the Nature Research Reporting Summary linked to this article.

## Data availability

The CIFAR10H[5] dataset is available at https://github.com/jcpeterson/cifar-10h. The raw data for NoisyCXR can be downloaded from the RSNA Pneumonia Detection Challenge[9], the detailed annotations per annotator for this challenge can be downloaded from here and the original labels from the NIH ChestXray dataset[3] can be found at the link. The labels used in our experiments are processed in the linked code repository https://github.com/microsoft/InnerEye-DeepLearning/tree/main/InnerEye-DataQuality. For further information on the acquisition and anonymisation of chest X-ray scans, and patients' informed consent, please see the online information[54]. The authors have not collected any data themselves as the study only used public, widely used anonymised datasets. The National Institutes of Health (NIH) Clinical Center was responsible for the release of the NIH dataset images (also used for the RSNA Pneumonia Detection Challenge). Senior Investigator: Ronald M. Summers, M.D., Ph.D., Senior Investigator of the Clinical Image Processing Service in the Imaging Biomarkers and Computer-Aided Diagnosis Laboratory of the NIH Clinical Center Radiology and Imaging Sciences Department. Additionally, the proposed study was reviewed by Microsoft Research Ethics Review Program's IRB, and met all the required ethical considerations outlined by the U.S. HHS Office for Human Research Protections, prior to initiation of the research (Reference IRB 10054, RCT 4018, 4375,4374). This was classified as non-human subjects research.

## Code availability

All the code for our label cleaning benchmarks, robust learning models, and experiments has been released in an open-source repository https://github.com/microsoft/InnerEye-DeepLearning/tree/main/InnerEye-DataQuality. The repository contains all the configuration files and library requirements for the reproducibility of experiments presented in this study.

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

## Acknowledgements

This work was funded by Microsoft Research Ltd (Cambridge, UK). The authors would also like to extend their thanks to Hannah Murfet for guidance offered as part the compliance review of the datasets used in this study.

## Author contributions

O.O., D.C.C., R.T., M.B. conceptualised and designed the benchmarks and methods presented in this manuscript. D.C.C., O.O, M.B., A.S. drafted the manuscript. D.C.C., O.O., M.B., J.A.V., R.T., S.B., M.L., A.N., B.G. revised the manuscript and gave critical feedback for important intellectual content. M.B., O.O., K.C.T., M.M. analysed the data. M.B., O.O., A.S., K.C.T., M.M. performed the experiments and statistical analysis. J.A.V. and A.N. secured funding. O.O., J.A.V., D.C.C., R.T. and A.N. supervised the project.

## Competing interests

The authors declare no competing interests.
