## [Peer Review File · Nature Communications]

Reviewers' Comments:

Reviewer #1:

Remarks to the Author:

This paper focused on active learning. As claimed by the authors, they proposed a novel learning framework termed active label cleansing, which aims to achieve a clean set of labels that can latter be used for benchmarking purposes. To this end, they proposed to rank instances according to estimated label correctness and labelling difficulty of each sample. Then, this paper conducted extensive experiments on two datasets, i.e., the CIFAR10H and the NoisyCXR. In experiments, the proposal could be 4 times more effective than a simple baseline in random selection.

The authors claimed that the traditional label noise learning approaches are insufficient, while I feel a little confused for the discussed reasons. First, the authors stated that there is no reliable method to determine whether the robust learning is effective. However, it seems that much of the previous approaches can adopt a clean validation set for evaluation. Second, the authors stated that label noise learning would like to disregard samples, but, in my opinion, it may not be true for the state-of-the-art methods. Here are two examples, among many others.

[1] Probabilistic End-to-end Noise Correction for Learning with Noisy Label. CVPR, 2019.

[2] DivideMix: Learning with Noisy Labels as Semi-supervised Learning. ICLR, 2020.

The objective of active label cleaning in Eq. 1 is reasonable, but I can hardly find it connection with the Algorithm 1 and the Eq. 2. It seems that the authors considered some kind of the dual form of the Eq. 1, such that the top-ranked samples should be relabelled. Then, by heuristics, the authors suggested the use of the cross entropy as well as the entropy term in designing the scoring functions. I would appreciate it if the authors could clarify the discussion of this part.

However, it seems that the classification model, i.e., $p(y | x; \theta)$, is trained and validated on the noisy dataset directly. In this case, the scoring function Φ in Eq. 2 would be inaccurate, i.e., both the noisiness and ambiguity would be low for all training sample, which is obviously detrimental for the ranking. Similar discussion is also mentioned by the authors: "the model rankings based on noisy validation metrics may be unreliable".

Reviewer #2:

Remarks to the Author:

General Comments:

In the manuscript "Active label cleaning: Improving dataset quality under resource constraints", Bernhard et al. propose a framework to perform label cleaning by scoring the data and prioritizing data that are more likely to be noisy for reannotation. The authors conducted a series of experiments with two datasets, CIFAR10H and NoisyCXR, evaluating the effectiveness of proposed cleaning approaches. Two critical aspects of the proposed method are an accurate way to determine relabelling priority and an approach to obtain clean labels on the identified high priority data. The author proposed a scoring function (Eq 2) combining cross-entropy from the normalized label counts to the predicted posterior and the entropy over posteriors.

Major Comments:

1. The accuracy of the scoring function is quite critical to the success of the proposed method. The author compared the proposed scoring function (Eq 2) with BALD in Table 2 with a 0.01 difference in performance (without showing the confidence interval). It will be helpful for the authors to mathematically discuss the difference between these functions, the pros and cons, and practically how a user should decide which scoring function to use.
2. L110: "each prioritised sample is reviewed sequentially by different annotators until a majority is formed among all the collected labels." The authors need to clarify and clearly define what does it mean by "a majority is formed". After all, there is a majority when there isn't a tie.
3. Based on the current algorithm, a data point is only selected once (L5 in Table 1) and removed from the set in L10 Table 1. Does that mean in that iteration, we have obtained the correct

annotation for the data point?

4. Related to the previous comment, the majority vote in data annotation is known to be suboptimal, as discussed in the following reference: Krause, Jonathan, et al. "Grader variability and the importance of reference standards for evaluating machine learning models for diabetic retinopathy." *Ophthalmology* 125.8 (2018): 1264-1272. Can the authors please discuss the implication of using a majority vote procedure here?

5. The authors proposed the data cleaning method by having an independent selector and classifier, where a selector is trained initially with the noisy train dataset, and the classifier is trained with the cleaned train dataset. In this approach, all the annotation budget B is used based on the selector. However, one can envision an approach of making this an iterative process by iterating between cleaning up the training dataset and training a better selector with the cleaner dataset. This iterative approach will likely lead to a more performant classifier with the same annotation budget, and potentially a higher quality cleaned dataset.

6. For Table 2, how does the setup of having Vanilla as selector and SSL as classifier perform? This result helps contextualize the performance of (4) in Table 2 by highlighting the degree of importance for using a Vanilla selector vs an advanced SSL selector.

7. From Fig 3, why don't the authors use the same set of algorithms running on both datasets? For example, there's SSL+coteaching in Fig 3a, but SSL and coteaching separately in Fig 3b.

8. Do the authors view the self-supervision with graph diffusion approach as a key contribution of this work? If not, I would recommend moving its related results to the supplementary as its inclusion makes the main story less clear. If this is indeed a key contribution of this work, please properly motivate and introduce the method in the main text and thoroughly evaluate the methods in the results section. For example, it's currently only in Fig 3b but not 3a.

9. For the implementation details in the supplementary material, are they for training both the classifier and selector? Or classifier only? Why not use the proposed scoring function (Eq 2) as the loss function for training the selector?

10. The authors discussed how this approach can be used for training and validation datasets but without discussing the test dataset. It will be helpful for the authors to discuss the test dataset applicability as well.

11. The annotation resources can be used to improve both the training set and the validation set. It would be great if the authors could discuss how the proposed method works when considering allocating annotation resources to both sets jointly.

Minor Comments:

1. What's the difference between the oracle and the minimal sampler? And why isn't oracle the minimal sampler?

2. It's not clear which is CIFAR10H and which is NoisyCXR in Supp Figure 3, and what's the respective dataset size. I assume it's the same as Fig 3 in the main manuscript, but please mark them to be clear.

3. For Figure 3 a, Vanilla (clean labels) seems to be doing way worse than oracle. In this setting, Vanilla is trained with the clean labels training set and used as a selector for prioritizing training set data for reannotation. Shouldn't Vanilla be performing close to optimal given that its performance is based on how well it can fit the training set?

4. L308: Supplementary Figure 2 is not about BALD. Do you mean Supp Figure 3?

Response Letter to Reviewers

Submission ID: NCOMMS-21-34829

Manuscript Title: “Active label cleaning for improved dataset quality under resource constraints”

Authors: Mélanie Bernhardt, Daniel C. Castro, Ryutaro Tanno, Anton Schwaighofer, Kerem C. Tezcan, Miguel Monteiro, Shruthi Bannur, Matthew P. Lungren, Aditya Nori, Ben Glocker, Javier Alvarez-Valle, Ozan Oktay

General comments

We would like to thank the reviewers for providing thoughtful and constructive feedback. They have both highlighted the novelty of the proposed active label cleaning approach. They also noted that the proposed methods have been studied in an extensive set of experiments. The reviewers pointed out some lack of clarity with respect to the “active” aspect of the framework, in which the model is updated throughout the cleaning process. They also asked for clarification with respect to the design and choice of the scoring function as well as the assumptions made for label aggregation.

We carefully took into account all of the given feedback, which has been incorporated into this revision. The main changes are summarised as follows:

- We have detailed how the selector model can be updated throughout the cleaning process, leveraging the set of corrected labels acquired so far (see Results section, lines 140–144)
- We have clarified the discussion around the scoring function, in particular how it differs from typical active learning objectives (see Results section and subsection titled “Sample scoring function impacts cleaning performance”)
- We have updated the discussion to highlight future work directions taking into account other label aggregation techniques, in particular in case of annotators with different levels of expertise (see Discussion section, lines 421–427).
- Our source code is now publicly available at <https://github.com/microsoft/InnerEye-DeepLearning/tree/main/InnerEye-DataQuality>
- The manuscript has been updated to adhere to the formatting guidelines.

Below we provide detailed responses to the comments from each reviewer, with localised references to the revised manuscript, where all changes are highlighted in blue.

Comments by Reviewer I

“ This paper focused on active learning. As claimed by the authors, they proposed a novel learning framework termed active label cleansing, which aims to achieve a clean set of labels that can latter be used for benchmarking purposes. To this end, they proposed to rank instances according to estimated label correctness and labelling difficulty of each sample. Then, this paper conducted extensive experiments on two datasets, i.e., the CIFAR10H and the NoisyCXR. In experiments, the proposal could be 4 times more effective than a simple baseline in random selection. ”

Authors: We thank the reviewer for the encouraging comments and the appreciation of the novelty of our work.

“ The authors claimed that the traditional label noise learning approaches are insufficient, while I feel a little confused for the discussed reasons. First, the authors stated that there is no reliable method to determine whether the robust learning is effective. However, it seems that much of the previous approaches can adopt a clean validation set for evaluation. Second, the authors stated that label noise learning would

like to disregard samples, but, in my opinion, it may not be true for the state-of-the-art methods. Here are two examples, among many others. [1] Probabilistic End-to-end Noise Correction for Learning with Noisy Label. CVPR, 2019. [2] DivideMix: Learning with Noisy Labels as Semi-supervised Learning. ICLR, 2020. ”

Authors: We would like to thank the reviewer for highlighting these aspects, which have been taken into account and clarified in the revised manuscript (see lines 46–61).

We claim that active learning (AL) and noise-robust learning (NRL) strategies can play a complementary role in improving the quality of datasets and also the performance of predictive models. In that regard, we propose to build upon the literature in both domains in a meaningful manner to clean noisy labels effectively.

In more detail, NRL approaches often cope with noise by inferring new labels [7] or disregarding samples [3] that could otherwise be highly informative or even be correctly labelled. However, models trained with these approaches can still learn biases from the noisy data, which may lead them to fail to identify incorrect labels, select already correct ones, or even introduce additional label noise via self-confirmation. Given these circumstances, NRL framework could benefit from the additional ground-truth information (e.g., active feedback from annotators) by incorporating the proposed labelling procedure into the learning framework, which in return yields better predictors as evidenced by the experimental results.

Lastly, NRL literature traditionally assumes that a clean evaluation set is available throughout the development process for model performance assessment. We clarify that this limitation refers to assessing the suitability of any trained model (noise-robust or otherwise) for a given real-world application, as clean evaluation labels are often unavailable in practice. This is in contrast to benchmarking algorithms in controlled experimental settings with curated datasets that may not align with the task at hand. We propose to address this challenge by actively collecting labels on the evaluation set, which can better guide NRL approaches.

“ The objective of active label cleaning in Eq. 1 is reasonable, but I can hardly find its connection with the Algorithm 1 and the Eq. 2. It seems that the authors considered some kind of the dual form of the Eq. 1, such that the top-ranked samples should be relabelled. Then, by heuristics, the authors suggested the use of the cross entropy as well as the entropy term in designing the scoring functions. I would appreciate it if the authors could clarify the discussion of this part. ”

Authors: Algorithm 1 is introduced to formulate the iterative label cleaning procedure by taking into account the objective in Eq. 1. In particular, lines 4 and 5 (Alg. 1) aim to minimise the number of incorrectly labelled samples whilst meeting the resource constraint. The algorithm is independent of the choice of scoring function (Eq. 2); for instance, it can be replaced with the BALD scoring function. This explanation is added into the revised manuscript.

In noise robust learning, the cross-entropy (CE) term has been commonly used in classification settings to identify and rank samples with noisy labels, such as in co-teaching [3], o2u-net [4] and truncated cross-entropy [5]. In these settings, model is used to estimate the noisy-labels and quantify their severity using a distance metric (CE) between predicted posteriors and existing labels. Similarly, in our framework the proposed CE term (Eq. 2) ranks samples using selector model in order to maximise the correctness term in Eq. 1. We have revised the manuscript to further clarify this link between our design choices and existing work in the literature (lines 127–129).

Similarly, the motivation for the use of entropy term in Eq. 2 is clarified (see lines 127–128), and we explained how it relates to the existing work in active learning (AL). In particular, uncertainty based AL methods [2, 6] focus on prioritising samples with high epistemic and low aleatoric uncertainty in choosing data points to be labelled next. Resource constraints (such as B in Eq. 1) and reduction of model uncertainty have been the two key main reasons for this choice of scoring function in these studies. Similarly, in our setting, we aim to prioritise samples with noisy label and low heteroscedastic aleatoric uncertainty (e.g., clear noise cases) as they require fewer labels for cleaning, where we expect better agreement between annotators in collected labels. In that regard, the entropy term is aimed to capture this quantity ($\mathbb{E}_{p(\theta|\mathcal{D})} H[p(\hat{y}|\mathbf{x}, \theta)]$), see Eq. 2 in [6]. As this uncertainty can be better estimated by marginalising it over θ distribution, we have conducted additional experiments using an ensemble of selector models. The results provided in Supp. Table 2 and Supp. Fig. 6 show that sample ranking

can be further improved by having better estimates of this uncertainty term if model ensembles can be computationally feasible to train. This information is also included in the main manuscript (lines 127–129) and also in supplementary material (Page 2, paragraph “Model ensembles yield better aleatoric uncertainty estimates”).

“ However, it seems that the classification model, i.e., $p(y|x; \theta)$, is trained and validated on the noisy dataset directly. In this case, the scoring function Φ in Eq. 2 would be inaccurate, i.e., both the noisiness and ambiguity would be low for all training sample, which is obviously detrimental for the ranking. Similar discussion is also mentioned by the authors: “the model rankings based on noisy validation metrics may be unreliable”. ”

Authors: We would like to thank the reviewer for the comment. The manuscript has been revised (lines 140–144) to further clarify this aspect as follows: We agree that noise in the training set is expected to affect the quality of the selector and we propose two solutions to tackle this challenge. First, noise robust learning techniques are introduced for the training of a sample selector (ranking). The simulation results show that noise-robust selectors (e.g. co-teaching or SSL) yield better sample ranking (Fig. 3) compared to a vanilla selector. Additionally, we also propose to regularly update the selector model and sample ranking (Φ) throughout the cleaning process, using the cleaned labels collected so far. This allows to iteratively improve the quality of the selector for more efficient sample selection in subsequent iterations. The effect of model updates is studied in detail in an ablation experiment presented in Supplementary Table 3.

Comments by Reviewer II

General Comments:

“ In the manuscript “Active label cleaning: Improving dataset quality under resource constraints”, Bernhardt et al. propose a framework to perform label cleaning by scoring the data and prioritizing data that are more likely to be noisy for reannotation. The authors conducted a series of experiments with two datasets, CIFAR10H and NoisyCXR, evaluating the effectiveness of proposed cleaning approaches. Two critical aspects of the proposed method are an accurate way to determine relabelling priority and an approach to obtain clean labels on the identified high priority data. The author proposed a scoring function (Eq 2) combining cross-entropy from the normalized label counts to the predicted posterior and the entropy over posteriors. ”

Authors: We would like to thank the reviewer for the comments to improve the clarity of the manuscript, and also for the ideas around additional experiments and future work.

Major Comments:

“ 1. The accuracy of the scoring function is quite critical to the success of the proposed method. The author compared the proposed scoring function (Eq 2) with BALD in Table 2 with a 0.01 difference in performance (without showing the confidence interval). It will be helpful for the authors to mathematically discuss the difference between these functions, the pros and cons, and practically how a user should decide which scoring function to use. ”

Authors: The scoring function in BALD [2] prioritises samples by taking into account the entropy between model weights and class posteriors associated with those samples. Specifically, the objective is $\Phi_{\text{BALD}}(\mathbf{x}; \mathcal{D}) = H(y|\mathbf{x}, \mathcal{D}) - \mathbb{E}_{p(\theta|\mathcal{D})}[H(y|\mathbf{x}, \theta)]$, where \mathbf{x} and \mathcal{D} are representing unlabelled data point and observed training set, respectively. With this objective, the highest scoring sample is prioritised to reduce this uncertainty and most often this corresponds to the samples closer to the decision boundary. From a model accuracy perspective, this approach shall yield improved performance (as shown in Table 2); however, this does not necessarily coincide with prioritisation and correction of incorrect labels. For example, as illustrated in Fig. 2, clear noise cases are not expected to be close to the decision boundary. On the other hand, Eq. (2), $\Phi(\mathbf{x}, \hat{\mathbf{l}}; \theta) = \text{CE}(\hat{\mathbf{l}}, p_{\theta}) - H(p_{\theta})$, prioritises the samples that fall under that

latter category (prioritising incorrect labels) hence optimising for label correctness directly. Hence, in Table 2 both selectors show improved accuracy before and after cleaning. However, the objective in BALD will focus on finding samples that may most affect the model parameters, not necessarily on cleaning as many samples as possible – as in the case of proposed scoring function. Hence, BALD does not perform as well for fixing incorrect labels as shown with grey and green curves in Supplementary Fig. 3a. We have clarified these points in the revised manuscript, see Results section and subsection “Sample scoring function impacts cleaning performance” (lines 314–317 and 323–328).

“ 2. L110: “each prioritised sample is reviewed sequentially by different annotators until a majority is formed among all the collected labels.” The authors need to clarify and clearly define what does it mean by “a majority is formed”. After all, there is a majority when there isn’t a tie. ”

Authors: After each labelling step by annotator, the incoming label is aggregated with the existing set of labels without differentiating between the annotators. In that regard, a standard majority class formation is performed as mentioned by the reviewer (*a majority is formed when there isn’t a tie*). However, each sample is required to contain more than one label for the majority rule to be applied. This has been clarified in the manuscript (line 117). The future work will explore modelling of each annotator and more optimal ways of label aggregation such as modelling annotator confusion matrices [1].

“ 3. Based on the current algorithm, a data point is only selected once (L5 in Table 1) and removed from the set in L10 Table 1. Does that mean in that iteration, we have obtained the correct annotation for the data point? ”

Authors: Yes, we make the assumption that once we have collected labels until majority, the final aggregate label becomes the best estimate of our ground-truth. In case this assumption does not hold, one could explore solutions where posterior beliefs are updated at each round of selection based on the number of already collected labels. In this way, we can avoid infinite sampling loops and use resources more effectively.

“ 4. Related to the previous comment, the majority vote in data annotation is known to be suboptimal, as discussed in the following reference: Krause, Jonathan, et al. “Grader variability and the importance of reference standards for evaluating machine learning models for diabetic retinopathy.” *Ophthalmology* 125.8 (2018): 1264-1272. Can the authors please discuss the implication of using a majority vote procedure here? ”

Authors: We agree with the reviewer that the majority vote can become sub-optimal when annotators have different levels of experience in labelling data samples. As a result, this can lead to ineffective use of resources hence the efficiency of label cleaning efforts. To address this, future work could explore sample selection and label assignment by taking into account the expertise of each annotator. We have expanded the discussion on this topic in the revised manuscript (see the Discussion section, lines 421–427).

More importantly, as discussed in the reference, there may not always be a certain ground-truth label but soft distribution across classes as in ambiguous cases described in the manuscript. This is quite commonly seen in healthcare related applications. In that regard, we have measured the total variation between sampled and ground truth distributions, the results are observed to be consistent with the simulation results in the main manuscript. We have now included this result in Supplementary Fig. 5.

“ 5. The authors proposed the data cleaning method by having an independent selector and classifier, where a selector is trained initially with the noisy train dataset, and the classifier is trained with the cleaned train dataset. In this approach, all the annotation budget B is used based on the selector. However, one can envision an approach of making this an iterative process by iterating between cleaning up the training dataset and training a better selector with the cleaner dataset. This iterative approach will likely lead to a more performant classifier with the same annotation budget, and potentially a higher quality cleaned dataset. ”

Authors: We agree with the reviewer’s comments, but the method proposed in the manuscript functions exactly as described by the reviewer. To further clarify, parameter θ (Table 1) corresponds to the model weights of the selector, which are updated in every b steps (line 13) with additional labels collected during the cleaning process. In return, this helps us to re-evaluate the scoring function for each remaining data point. The impact of the frequency at which the selector model is updated throughout the cleaning process has been further explored in Supplementary Table 3. As pointed out in the review comments, this analysis showed that all selectors benefited from being updated regularly with the set of labels acquired so far. In the revised manuscript, this has been further clarified (see lines 140–144) and we would like to thank to the reviewer for pointing this out. The corresponding source code can be found at¹.

“ 6. For Table 2, how does the setup of having Vanilla as selector and SSL as classifier perform? This result helps contextualize the performance of (4) in Table 2 by highlighting the degree of importance for using a Vanilla selector vs an advanced SSL selector. ”

Authors: We have run the additional experiment as suggested by the reviewer - Table 2 in the manuscript is now extended (see row 5, Table 2). Results show that using a better selector enables further performance improvements compared to using a vanilla selector, independently of the choice of classifier.

“ 7. From Fig 3, why don’t the authors use the same set of algorithms running on both datasets? For example, there’s SSL+coteaching in Fig 3a, but SSL and coteaching separately in Fig 3b. ”

Authors: For both datasets, we show the performance of different selectors, including random, oracle, vanilla and co-teaching. As pointed out by the reviewer, both graphs differ in terms of which flavour of SSL fine-tuning they depict, thus the two figures do not necessarily differ from each other.

Our preliminary experiments on chest X-ray data showed that it was not sufficient to train a linear classifier on top of a frozen image encoder for this task. The complexity of the chest X-ray task simply requires more model capacity at fine-tuning time, compared to the CIFAR10H case. Hence, we chose to show the performance of SSL + coteaching for chest X-rays, whereas for CIFAR10H we show that a simple linear classifier on top of the pre-trained encoder was already outperforming all other selectors. Therefore, we preferred the use of simpler models whenever it is applicable depending on the complexity of the task, and pretrained SSL weights improves the results for both datasets. This is further highlighted in the manuscript (lines 287–291).

“ 8. Do the authors view the self-supervision with graph diffusion approach as a key contribution of this work? If not, I would recommend moving its related results to the supplementary as its inclusion makes the main story less clear. If this is indeed a key contribution of this work, please properly motivate and introduce the method in the main text and thoroughly evaluate the methods in the results section. For example, it’s currently only in Fig 3b but not 3a. ”

Authors: The proposed graph-based approach was included as a comparison to a simple linear classifier based approach, and it was not one of our main contributions. As we agree with the reviewer’s comment that it may make the main story less clear, we have opted for removing the graph diffusion approach from the manuscript, as suggested by the reviewer.

“ 9. For the implementation details in the supplementary material, are they for training both the classifier and selector? Or classifier only? Why not use the proposed scoring function (Eq 2) as the loss function for training the selector? ”

Authors: The classifier and selector are trained with the same configuration but they differed in terms of the labels used to train them. This information is now further highlighted in the text (see Implementation details, in Supplementary Material, lines 140–142 and 180–182).

¹https://github.com/microsoft/InnerEye-DeepLearning/blob/main/InnerEye-DataQuality/InnerEyeDataQuality/selection/selectors/label_based.py

We agree with the reviewer that one could use the entropy term in Eq. 2 to train selector models in addition to the log-likelihood term. It has been commonly used in the literature to regularise models to avoid over-fitting to target distributions, which could serve as another means of label noise robustness. However, in our setting the uncertainty term is mainly meant for the ranking of samples to determine the order in which they will be labelled instead of a general regularisation at training time.

“ 10. The authors discussed how this approach can be used for training and validation datasets but without discussing the test dataset. It will be helpful for the authors to discuss the test dataset applicability as well. ”

Authors: We would like to clarify that ‘evaluation set’, as defined in the manuscript, means generically validation and/or test data. In that regard, our approach can be equally applied to rank and prioritise samples in test dataset.

“ 11. The annotation resources can be used to improve both the training set and the validation set. It would be great if the authors could discuss how the proposed method works when considering allocating annotation resources to both sets jointly. ”

Authors: The method can be applied without changes by pooling any combination of training and evaluation samples. Note that typical concerns about information leakage leading to overoptimistic performance assessment do not apply in this case, as cleaning evaluation labels is generally expected to *reduce* downstream evaluation bias irrespective of what data is used to train the selector.

In practice, when employing a classifier that is already highly robust to noise, the accuracy gains from cleaning the training data can be only marginal (see e.g. SSL in Table 2). On the other hand, our paper mainly advocates for cleaning the evaluation set, as this will more reliably estimate predictive performance regardless of the chosen model. We therefore advise prioritising the cleaning of evaluation data up to an acceptable level, before applying any remaining budget to clean the training set. We have now included this explicit recommendation (lines 351–355).

“ Minor Comments:

- 1. What’s the difference between the oracle and the minimal sampler? And why isn’t oracle the minimal sampler?*
- 2. It’s not clear which is CIFAR10H and which is NoisyCXR in Supp Figure 3, and what’s the respective dataset size. I assume it’s the same as Fig 3 in the main manuscript, but please mark them to be clear.*
- 3. For Figure 3 a, Vanilla (clean labels) seems to be doing way worse than oracle. In this setting, Vanilla is trained with the clean labels training set and used as a selector for prioritizing training set data for reannotation. Shouldn’t Vanilla be performing close to optimal given that its performance is based on how well it can fit the training set?*
- 4. L308: Supplementary Figure 2 is not about BALD. Do you mean Supp Figure 3?*

”

Authors:

- 1. The oracle has access to the true label distribution, and it samples labels from this distribution during the relabelling process. It may therefore sample labels that may differ from the actual majority class. The minimal sampler on the other hand always returns the ground-truth label without emulating the inherent ambiguity of a given sample (unlike the Oracle). This has been clarified in the manuscript (lines 231–238).*
- 2. Indeed, the order is the same as in the main manuscript. The caption has been updated in the revised version of the supplementary material.*

3. The chest X-ray classification is inherently a more challenging problem; thus the corresponding selector models do not perform equally well on this task as in CIFAR10H. In other words, even when training the selector on clean data, the posteriors estimated by the underlying model might not always be accurate. This is how we reason about the wider gap between the oracle and the selector trained on clean data. Therefore, the selector trained on clean data constitutes here a “model upper bound”. This has been precised in the caption of Figure 3.
4. We thank the reviewer for spotting this typo, the reference is updated in the revised manuscript.

References

- [1] Tanno, R. Saeedi, A., Sankaranarayanan, S., Alexander, D. C. & Silberman, N. Learning from noisy labels by regularized estimation of annotator confusion. In *Proceedings of the IEEE/CVF Conference on Computer Vision and Pattern Recognition (CVPR)*, 11244–11253 (2019).
- [2] Houthby, N., Huszár, F., Ghahramani, Z. & Lengyel, M. Bayesian active learning for classification and preference learning (2011). URL <https://arxiv.org/abs/1112.5745>.
- [3] Han, B. *et al.* Co-teaching: Robust training of deep neural networks with extremely noisy labels. In *Advances in Neural Information Processing Systems*, 8527–8537 (2018).
- [4] Huang, J., Qu, L., Jia, R. & Zhao, B. O2U-Net: A simple noisy label detection approach for deep neural networks. In *Proceedings of the IEEE/CVF International Conference on Computer Vision*, 3326–3334 (2019).
- [5] Zhang, Z. & Sabuncu, M. Generalized cross entropy loss for training deep neural networks with noisy labels. In *Advances in Neural Information Processing Systems 31*, 8778–8788 (2018).
- [6] Kirsch, A., van Amersfoort, J. & Gal, Y. BatchBALD: Efficient and diverse batch acquisition for deep bayesian active learning (2019). [arXiv:1906.08158](https://arxiv.org/abs/1906.08158).
- [7] Junnan Li, S. C. H. H., Richard Socher. DivideMix: Learning with noisy labels as semi-supervised learning. In *International Conference on Learning Representations* (2020).

Reviewers' Comments:

Reviewer #1:

Remarks to the Author:

Many thanks for the revision. The authors not only address my concerns and also make their paper much clear. In high level, the authors propose a general framework that could combine active learning and noise-robust learning. However, I still have some little confusion about the paper, and I would appreciate it if the authors could discuss about the following question.

The authors claimed in the new version that NRL methods can still learn bias from noisy data. However, since the proposal use these methods in scoring, there exist some samples with incorrect labels but high confidence. Since these samples will not be selected for relabelling, it means that the selected samples contain bias and the iterative cleansing procedure may not correct this bias throughout the training. I wonder which part in the experiments can verified that this bias could be properly corrected. If not, maybe the attached reference will be helpful.

Cheng, Jiacheng, et al. "Learning with bounded instance and label-dependent label noise." International Conference on Machine Learning. PMLR, 2020.

Reviewer #2:

Remarks to the Author:

Thanks to the authors for addressing the previous comments.

For the discussion in #10 and #11 on using the proposed approach to rank and prioritize samples in the test dataset, it would be great if the authors could discuss and elaborate more about how using the proposed approach won't lead to an over-optimistic performance in the discussion section. Typically, when designing a test set, we would like the test set to be completely independent in terms of both data source and the annotation process for good data practice hygiene. It's not clear whether using the proposed approach to perform label cleaning on the test set breaks the independence or not, as the same selector is used for both cleaning the train/validation set and the test set.

Response Letter to Reviewers

Submission ID: NCOMMS-21-34829B

Manuscript Title: “Active label cleaning for improved dataset quality under resource constraints”

Authors: Mélanie Bernhardt, Daniel C. Castro, Ryutaro Tanno, Anton Schwaighofer, Kerem C. Tezcan, Miguel Monteiro, Shruthi Bannur, Matthew P. Lungren, Aditya Nori, Ben Glocker, Javier Alvarez-Valle, Ozan Oktay

General comments

We would like to thank the reviewers for providing thoughtful and constructive feedback. They have both highlighted the improved clarity of the revised manuscript, and we are pleased to see that most of the initial concerns were addressed in the previous revision. In their latest feedback, the reviewers pointed out some lack of clarity with respect to (I) the potential limitations of the proposed approach in terms of the noise characteristics, and (II) guidelines to ensure the independence of the test set and minimise any potential information leakage between the data splits. We have carefully taken into account these two comments and aimed to address them in this revision. The main changes are summarised as follows:

- We have clarified the underlying assumptions we make on the noise characteristics and the associated conditions required for data-driven approaches to operate as expected. Additionally, the suggested reference is included in the discussion section to further corroborate the value of merging both AL and NRL methods to form expert-distilled samples and cope with consistent mislabels (see Discussion section, lines 423–451).
- We have revised the discussion in the main manuscript and provided a set of recommendations to follow in curating test set labels. In particular, we have taken into account the independence of test set and any potential biases that may arise due to the choice of selector model (see Discussion section, lines 401–411).

Below we provide detailed responses to the comments from each reviewer, with localised references to the revised manuscript, where all changes are highlighted in blue.

Comments by Reviewer I

“ Many thanks for the revision. The authors not only address my concerns and also make their paper much clear. In high level, the authors propose a general framework that could combine active learning and noise-robust learning. ”

Authors: We would like to thank the reviewer for the insightful comments and appreciation of the clarity of the revised manuscript.

“ However, I still have some little confusion about the paper, and I would appreciate it if the authors could discuss about the following question. The authors claimed in the new version that NRL methods can still learn bias from noisy data. However, since the proposal use these methods in scoring, there exist some samples with incorrect labels but high confidence. Since these samples will not be selected for relabelling, it means that the selected samples contain bias and the iterative cleansing procedure may not correct this bias throughout the training. I wonder which part in the experiments can verified that this bias could be properly corrected. If not, maybe the attached reference will be helpful.

Cheng, Jiacheng, et al. “Learning with bounded instance and label-dependent label noise.” International Conference on Machine Learning. PMLR, 2020. ”

Abbreviations - AL: Active Learning, **NRL:** Noise-robust Learning

Authors: The reviewer has highlighted an important aspect of the problem, and it needs to be taken into account when data-driven approaches are utilised to handle noisy labels that occur in a consistent manner. For that reason, we have further clarified the underlying assumptions for the proposed approach to function as expected, and analysed how different approaches (e.g., AL+NRL, NRL, Random) can handle such extreme label noise characteristics.

[Cheng et al. ICML’20] outlines the common noise assumptions that data-driven approaches (NRL and AL+NRL) quite often rely on, including the bounded noise rate assumption, where on average there are more correct labels than wrong in a given dataset. Under extreme conditions where this assumption may not hold, random selection can become the preferred method over data-driven approaches to better handle such consistent mislabels and correct them. Despite this fact, our approach does not flip the already correct labels assuming that manual labellers provide i.i.d. samples from the true data distribution \mathcal{D}^* and a sufficient number of labels are collected. From this perspective, we are actually converging towards the true marginal data distribution, address certain subset of label inconsistencies, and optimise for the objective given in Eqn. (1).

More importantly, when the above two conditions hold true (e.g., bounded noise rate), as often the case in real-world applications, then the AL component of the proposed framework can potentially address such consistent mislabels, which is further corroborated by Tables 1 and 2 in [Cheng et al. ICML’20]. In detail, AL can establish a distilled set of expert labels to tackle this challenge instead of solely relying on self-distillation or hallucination of the true labels as in NRL methods. To further extend our methodology along these lines, one could treat the newly acquired labels as expert-distilled samples, and assign them higher weighting in posterior updates in the proposed iterative framework.

Lastly, it is important to note that the proposed label cleaning procedure is not a fully automated process. It is a human-in-the-loop system; as such the prioritised samples and their collected labels can be monitored over multiple iterations. Therefore, from a practical point-of-view, the process can be intervened on whenever the underlying assumption may not hold, and there is a concern around group-specific mislabels in the dataset. **The manuscript has been updated to reflect this discussion, please see lines 423–451 in the Discussion section.**

Comments by Reviewer II

“ Thanks to the authors for addressing the previous comments. For the discussion in #10 and #11 on using the proposed approach to rank and prioritize samples in the test dataset, it would be great if the authors could discuss and elaborate more about how using the proposed approach won’t lead to an over-optimistic performance in the discussion section. Typically, when designing a test set, we would like the test set to be completely independent in terms of both data source and the annotation process for good data practice hygiene. It’s not clear whether using the proposed approach to perform label cleaning on the test set breaks the independence or not, as the same selector is used for both cleaning the train/validation set and the test set. ”

Authors: We thank the reviewer for the valuable and constructive feedback provided in the revision rounds. After further reflecting on the discussion in #10 and #11, we have prepared a set of guidelines to ensure good data practice hygiene as pointed out by the reviewer.

- Selector models (θ) utilised for sample prioritisation and label cleaning should not be used for classification and evaluation purposes on the test set, which we strictly avoid in the experiments. These models shall be discarded once the label cleaning procedure is complete.
- If the objective is to curate the test set, then we recommend that selector models are trained and applied only on the test set in order to avoid any potential information leakage between the train and test splits.
- Lastly, there could be an inherent model bias in ranking of the samples due to use of a data-driven approach, which is independent of the potential information leakage between train and set splits. As a remedy to this, the posteriors (Eqn. 2) can be estimated with an ensemble of models with different formulations and inductive biases. In that regard, the framework does not make any assumption on the family of functions (θ) that can be used for label cleaning.

These recommendations have now been included in the manuscript, in lines 401–411 in the Discussion section.

Reviewers' Comments:

Reviewer #1:

Remarks to the Author:

The rebuttal has addressed all my concerns. I have no further comments.

Reviewer #2:

Remarks to the Author:

The authors have fully addressed my comments, thanks for revising the manuscript to incorporate the feedback.